# Sound-Event Detection of Water-Usage Activities Using Transfer Learning

**DOI:** 10.3390/s24010022

**Published:** 2023-12-19

**Authors:** Seung Ho Hyun

**Affiliations:** School of Electrical Engineering, University of Ulsan, Ulsan 44610, Republic of Korea; takeitez@ulsan.ac.kr; Tel.: +82-52-259-1277

**Keywords:** sound-event detection, transfer learning, water-usage activities, YAMNet

## Abstract

In this paper, a sound event detection method is proposed for estimating three types of bathroom activities—showering, flushing, and faucet usage—based on the sounds of water usage in the bathroom. The proposed approach has a two-stage structure. First, the general sound classification network YAMNet is utilized to determine the existence of a general water sound; if the input data contains water sounds, W-YAMNet, a modified network of YAMNet, is then triggered to identify the specific activity. W-YAMNet is designed to accommodate the acoustic characteristics of each bathroom. In training W-YAMNet, the transfer learning method is applied to utilize the advantages of YAMNet and to address its limitations. Various parameters, including the length of the audio clip, were experimentally analyzed to identify trends and suitable values. The proposed method is implemented in a Raspberry-Pi-based edge computer to ensure privacy protection. Applying this methodology to 10-min segments of continuous audio data yielded promising results. However, the accuracy could still be further enhanced, and the potential for utilizing the data obtained through this approach in assessing the health and safety of elderly individuals living alone remains a topic for future investigation.

## 1. Introduction

In the evolving realm of smart-home technologies, there is a growing emphasis on ensuring the safety and well-being of elderly individuals living alone [1]. Bathrooms, being integral to daily routines, also pose inherent risks, making them a point of heightened concern. Traditional wearable sensors, while beneficial, have limitations such as frequent recharging requirements and discomfort during extended use [2]. Furthermore, considering personal privacy, video-based monitoring systems are not suitable for these intimate spaces [3]. Additionally, there is currently a lack of readily available, easy-to-install IoT flow sensors for monitoring the water usage at each outlet. A non-intrusive sensor capable of seamlessly detecting and recording bathroom water-usage activities could be invaluable for monitoring the safety of elderly individuals living alone and for evaluating their health status [4,5].

Leveraging the potential of sound event detection (SED), a technology capable of analyzing non-verbal acoustic signals to infer the circumstances producing the sound, our research team has proposed a system which utilizes an artificial intelligence classification model [5]. Implemented on an edge computing device, this system is designed to automatically detect water-usage activities in the bathroom and subsequently record and upload the activity log onto the cloud. The conceptual framework of this system is illustrated in Figure 1. Water-usage activities in the bathroom were categorized as “Flush”, “Shower”, and “Faucet”, while sounds originating from non-water-related activities were categorized as “Others”. Taking into account the unique acoustic characteristics arising from variations in plumbing, faucet structures, and floor materials in each bathroom, and recognizing that the edge device will receive sounds exclusively from its installation location, sounds were recorded directly in the intended installation environment. This dataset of recorded sounds served as the training set for the AI model aimed at detecting water-usage activities.

However, there were noticeable challenges. The accuracy in distinguishing between “Shower” and “Faucet” sounds was not consistently satisfactory across different bathrooms [5]. Additionally, the model faced difficulties in discerning sounds unrelated to water usage, particularly due to the limited diversity within the “Others” class in our training data. This highlighted the need for further refinement and enrichment of the dataset to improve the overall performance of the system.

Transfer learning with convolutional neural networks (CNNs) has become pivotal in sound classification [6,7]. Instead of training deep networks from scratch, which is resource-intensive, researchers use CNNs pre-trained on large image datasets like ImageNet [8,9,10,11]. These networks, though rooted in visual domains, have layers with generic features adaptable to auditory tasks. With fine-tuning on target sound data, they enable the rapid development of sound models, even with limited auditory data. Google’s AudioSet [12], an extensive dataset comprising over 2 million annotated audio clips from YouTube, led to the development of YAMNet—a deep neural network built on the Mobilenet_v1 architecture [13,14]. Optimized for on-device use, YAMNet identifies a wide range of sound events, serving various audio applications [15].

In this study, we utilized YAMNet as a pre-trained model, employing sound recordings of three types of water-usage activities from the designated bathroom as the dataset for transfer learning. We trained a model to detect three specific water-usage activities: flushing, showering, and faucet usage. This model was then implemented on an edge device, and its performance was evaluated. Before distinguishing between the three types of water sounds, we combined this approach with a method that detects non-water sounds as “Others” using YAMNet, which was originally trained on AudioSet.

## 2. Materials and Methods

### 2.1. Sound-Event Detection by YAMNet

The aim of this paper is to classify real-time water-usage activities like showering (SH), flushing (FL), and faucet usage (FA) through an analysis of sounds in a bathroom. This falls within the area of sound-event detection (SED), which has made notable advancements thanks to the application of deep learning techniques. Convolutional neural networks (CNNs) are commonly employed as deep learning models due to their capacity to effectively extract meaningful features from data. Raw audio data is not suitable as a direct input for CNNs; therefore, appropriate transformations are needed. The Mel spectrogram is widely acknowledged as the most suitable input format for this purpose. A Mel spectrogram translates acoustic signals into a visual representation that depicts sound intensity in Mel-frequency bands over time, and it is widely used in audio data processing. In essence, in other words, with the Mel spectrogram transforms applied, SED is shifted from a problem of distinguishing the inherent acoustic characteristics in sound to a challenge of distinguishing the visual features of the Mel spectrogram [16,17,18].

In the realm of audio processing and SED, Google has taken on a pioneering role by introducing the comprehensive AudioSet dataset. AudioSet encompasses a vast collection of manually annotated audio events, extracted from over 2 million 10-s YouTube video clips, effectively categorizing them into 521 distinct sound-event classes. They have also released YAMNet, a deep neural network model specifically designed for the event recognition of non-verbal sounds. Built upon the efficient Mobilenet_v1 depthwise-separable convolution architecture, YAMNet is not just optimized for on-device applications but also exhibits superior performance in recognizing a broad spectrum of sound events [9,12].

However, YAMNet exhibits both positive and negative aspects when used for our purpose. Looking first at the positive aspects: YAMNet has been trained on a vast amount of data and can identify sounds related to water with a fairly high probability. This model assigns the probabilities of the input sound matching one of the pre-defined 521 individual classes, some of which are related to water sounds. Actually, when the water sounds collected in the bathrooms for this study, i.e., SH, FL, and FA, are input into YAMNet, the sum of the probabilities related to the water sounds is calculated to be about 83%. Of course, within this 83%, there are also other answers like “bath tube filling or washing”, “drip”, “liquid”, and more. However, the significance of “an 83% likelihood of being related to water use” is that it provides clarity on whether or not the input sound is related to water.

However, there are two noteworthy limitations to consider. The first limitation is that the classes provided by YAMNet may not encompass the specific behaviors required for practical applications. For instance, the behavior “Shower”, which is one focus of this study, is not included in the classes provided by YAMNet. The second limitation is that it may not adequately reflect the specific characteristics of the individual environments in which the sound originates. YAMNet has been trained on a vast dataset collected from various sources. In essence, while it adeptly captures common and general features, it might encounter difficulties in representing the unique characteristics of sound origins. The sounds examined in this study, i.e., SH, FL, and FA, are fundamentally associated with water usage, and each has its unique characteristics. However, even within the faucet sound category, significant variations can arise due to factors like bathroom structure, sink materials, water pressure, and plumbing configurations in different households. Differences resulting from such diversities may not be adequately represented in YAMNet’s training data. At the beginning of our research for this paper, we applied the sound data aquired from three different bathrooms with distinct environments to YAMNet directly. According to our experiments, when attempting to classify the sounds using YAMNet, only 65% of the 398 flushing sounds and 34% of the 410 faucet usage sounds were accurately estimated. In other words, even though YAMNet can detect the basic characteristics of water sounds, it is far from suitable for deployment in individual bathrooms. Therefore, in this study, we propose a method that leverages the positive aspects of YAMNet and addresses its limitations, which will be explained in the following subsection.

### 2.2. Water-Sound Activity Classifier: W-YAMNet

In a bathroom, various sounds besides water-usage sounds can occur, such as the sound of slippers, human voices, electric shavers, music, and others. If our deep learning model has been trained only on the water sounds of SH, FL, and FA, and has three output classes, it will classify sounds other than water-usage sounds as one of the three water-sound classes. To distinguish these non-water-related sounds from water sounds, one possible approach is to define and train separate classes for sounds other than water sounds. However, in this case, it is hardly expected that various sounds originating from different causes will share common characteristics. Therefore, grouping them all into one class may not be effective. On the other hand, specifying each sound cause as a separate class may result in an excessively large number of classes, and the collection of training data for all of these classes could be severely restricted. Therefore, it is reasonable to begin the process of classification by first determining whether the sounds are related to water usage. Once it is confirmed that the sounds are indeed related to water usage, the classifier can proceed to distinguish the specific class among SH, FL, and FA. In this study, we introduce a classifier that consists of two distinct phases: the “water-usage sound determination stage” and the “specific classification stage”.

In the first stage, we aimed to leverage the advantageous capability of YAMNet as described earlier, that is, its ability to discern the fundamental characteristics of water-related sounds. When presented with an input sound clip of a specified duration, YAMNet provides the probability that the sound corresponds to one of the 521 classes. Consequently, by combining the probabilities of all classes related to water usage within these 521 classes, we can evaluate the probability that the input sound is associated with water usage. For convenience, we refer to the sum of these probabilities as the “water-related-sound score (WRS score)”. The detailed explanation of how to calculate the WRS score is provided in Section 3.2.

Once the WRS score indicates a water-related sound, the second-stage process is initiated to determine whether the sound corresponds to SH, FL, or FA. Here, again, YAMNet is employed. However, with the reduction in output classes from 521 to 3, structural adjustments are necessary, and, as previously mentioned, new partial training is required to incorporate the detailed characteristics of the bathroom where the system is to be applied. In this, transfer learning proves to be effective.

Transfer learning is a machine-learning technique where a model developed for one task is reused as the starting point for a model on a second task. Essentially, it harnesses the knowledge gained from training on one problem to improve the performance on a related problem [14,18]. Given the substantial computational cost and data requirements for training deep models from scratch, utilizing pre-trained models such as YAMNet, which has been trained on expansive audio datasets, is a viable strategy. YAMNet serves as the foundational architecture (or the pre-trained model) from which the classifier layer is adapted or retrained to cater to specific application needs. By retraining only the classifier layer and leveraging the learned features of YAMNet, one can achieve effective and efficient sound classification tailored to specific tasks without the need for extensive data or computational resources [19]. Researchers have utilized a retrained YAMNet to accurately identify a range of sounds from common household activities like the hum of a fridge to industrial machinery anomalies and urban noises like car honks. By selectively retraining the last few layers of YAMNet, high precision was achieved in distinguishing these diverse sounds, offering potential applications in smart-home setups, care monitoring of the elderly, machinery malfunction detection, and urban noise pollution studies.

In this study, we defined the transfer learning model of YAMNet as Water-YAMNet, which is abbreviated to W-YAMNet. The original YAMNet has 1024 nodes in its final hidden layer, and the output layer consists of 521 output nodes, each corresponding to a different class. In W-YAMNet, YAMNet’s output layer was replaced with another hidden layer containing 512 nodes, and an output layer with 3 nodes corresponding to SH, FL, and FA was added. As a result, 1024 × 521 weights were removed, and 1024 × 512 + 512 × 3 new weights were introduced. There is no straightforward analytical method to figure out the number of nodes in the added layer or, in other words, the number of weights that require training. Nevertheless, after running multiple simulations, we settled on using 512 nodes, which is half the number of nodes in YAMNet’s last hidden layer. These newly added weights are trained to incorporate the acoustic information collected from each bathroom, allowing the model to adapt to characteristics such as the bathroom’s structure, floor material, and water pressure, as well as user habits.

In W-YAMNet, the requirement to modify trainable parameters is notably reduced compared with the training of an entirely novel model. As a consequence, the data volume necessary for training is reduced too. This efficiency is attributed to the substantial pre-existing knowledge of water-related sounds that YAMNet has acquired. Consequently, W-YAMNet primarily focuses on the integration of the distinctive attributes associated with the specific bathroom under consideration. Considering the need to collect data as quickly as possible in individuals’ living spaces, this can be a significant advantage.

The relationship between YAMNet and W-YAMNet is illustrated in Figure 2. In this figure, the dashed encircled area represents the shared portion between YAMNet and W-YAMNet. YAMNet’s output layer, consisting of 521 nodes, has been replaced in W-YAMNet with a hidden layer of 512 nodes and an additional output layer containing 3 nodes. The complete process explained above can be depicted in the flowchart illustrated in Figure 3.

## 3. Experimental Results

### 3.1. Overview of Experiments

In this section, we explain the experimental results of classifying bathroom water sounds into the three classes of FL, FA and SH, using the previously mentioned YAMNet and W-YAMNet models. For data acquisition and classification, the Raspberry Pi 4 (Cortex-A72, 64-bit SoC @ 1.5 GHz, 4 GB) edge computing device shown in Figure 4a was utilized. An omnidirectional condenser microphone (CMA-4544PF) from CUI Inc., Tualatin, OR, USA, was connected to the Raspberry Pi 4. YAMNet and W-YAMNet were deployed on the edge computer, and all incoming sound data were processed and evaluated here. Only the water-usage detection results and timestamp information were stored on the cloud computer as depicted in Figure 4b [20]. This was to protect personal privacy, as water sounds can sometimes have the potential to expose personal matters.

The data collection for training and performance evaluation took place in the bathrooms of three different houses. Each bathroom has a distinct toilet structure, floor material, water pressure, and showerhead, resulting in distinct characteristics for each water-usage sound. In training and testing, the sound recording and labeling were conducted by the researcher and an assistant who directly manipulated the water-related devices. Data were collected at a sampling rate of 16 kHz and with a 16-bit resolution. Both the training and evaluation datasets were constructed from data obtained on the same day and used for training W-YAMNet. The training of the W-YAMNet model was performed using Google-provided servers within the Colab (Google Colaboratory) environment using the TensorFlow library [21].

After the training was completed, the W-YAMNet classifier was evaluated using new data collected on different days from the training data. Several factors influenced the classification performances, which will be explained in more detail in the remaining part of this section.

### 3.2. Water-Related Sound Score Using the YAMNet Output

In our study, we analyzed audio recordings taken in three different bathroom settings. We captured a total of 1224 audio clips, each lasting 3 s, while various water-usage activities were taking place. These audio clips were then run through the YAMNet model. From the 521 possible categories that the model could identify, we compiled the average values for the top 10 categories with the strongest predictive scores, which can be found in Table 1.

Among the 521 classes, 8 (highlighted in gray in Table 1) were distinctly water-related: “Water”, “Water tap, faucet”, “Sink (filling or washing)”, “Toilet flush”, “Bathtub (filling or washing)”, “Liquid”, “Fill (with liquid)”, and “Drip”. The output scores from YAMNet for these eight classes were collectively defined as the water-related sound score (WRS score). If the value of the WRS score exceeds a certain threshold, it can be determined that the sound input to the microphone corresponds to a water-usage activity. If the threshold value is set high, the ability to correctly identify water sounds as water sounds, i.e., the sensitivity, is likely to decrease, while the ability to correctly distinguish non-water sounds, i.e., the specificity, is expected to increase. Conversely, lowering the threshold value is likely to yield the opposite results.

The threshold value can only be determined through trial and error. For our purposes, it is necessary to evaluate WRS scores for various bathroom noises that are not water-related sounds. To establish this threshold value, we collected 734 clips from each bathroom containing sounds other than water-related noises. These clips encompass a variety of sounds, including hairdryers, electric razors, conversations, music, and more.

Figure 5 shows the WRS score histogram in each case for the 1224 water-usage sound clips (left) and for clips containing other sounds only (right). The figure clearly demonstrates that the WRS scores for water-usage sound clips are notably higher compared with those of other sound clips. In the case of other sounds, the majority of the WRS scores fall within the range of 0–0.1, and there are no scores that surpass the threshold of 0.25. From this observation, a threshold of 0.1 was set for the WRS score results, with a sensitivity of 0.917 and a specificity of 0.920, demonstrating a well-balanced detection of water-usage sounds.

However, for the objectives of this study, a greater emphasis was placed on reducing false positives. Hence, a threshold value of 0.25 was adopted, categorizing any clip with WRS score exceeding this threshold as a water sound, which would subsequently be classified. With this threshold value, a sensitivity of 84.2% and a specificity of 99.6% were attained, which was considered suitable for the objective of minimizing false positives.

### 3.3. Prediction Performance of W-YAMNet

#### 3.3.1. Training of W-YAMNet

As previously mentioned, the YAMNet model was trained through transfer learning, and for analysis and comparison, we separately built three different models using sounds from three different bathrooms labeled T01, T02, and T03. The datasets for training and validation for each model are tabulated in Table 2. The dataset which is immediately input into YAMNet is referred to as a “frame”. The length of a frame is 0.96 s; therefore, the training and validation of W-YAMNet also used frames of the same length.

Since the sound data are divided into clips, each clip should be further subdivided into an appropriate number of frames. An N-second sound clip is divided into 1-s frames, with each frame overlapping the previous one with a hopping interval of 0.5 s, resulting in a total of 2 N frames, among which the last one contains data only in its front half. The data quantities presented in Table 2 are expressed in terms of the number of sound clips, each of which has a length of 3 s.

#### 3.3.2. Frame-Level and Clip-Level Classification and Clip Length

Both YAMNet and W-YAMNet are designed to provide input and output at the frame level. However, the characteristics of sounds generated by target activities can change momentarily. Additionally, noises that can occur within a bathroom, such as the sound of slippers, objects being dropped, coughing, etc., may mix with water sounds. These sounds can lead to classification errors; however, they often occur and disappear in very short moments rather than lasting for extended periods. Hence, when assessing the results at the frame level, there is a greater chance of being impacted by errors originating from these phenomena. Therefore, in this study, the unit of classification was defined as a clip, not a frame; in other words, the results of frames included within a clip were averaged for classification. We also conducted experimental assessments of the classifier’s performance by gradually reducing the clip length by 1 s, starting from 5 s, and, of course, we retrained W-YAMNet to align with the new clip length.

Table 3 provides a comparison between the frame-level and clip-level results for various clip lengths for each bathroom, which allows us to assess the classification accuracy of W-YAMNet when the clip length is altered. For instance, with a 5-s clip length, the F1 score for the frame-level classification is 0.950, while the clip-level F1 score is 1.00, indicating that the classifications were perfect. For different clip lengths, it can also be observed that clip-level classification is more accurate compared with frame-level classification. This could be attributed to issues in training, but it is more likely that momentary noise or factors related to the characteristics of each bathroom, such as water pressure, have a greater impact on these misclassifications. As is evident in this table, this trend holds consistently across all bathrooms, highlighting the effectiveness of clip-level classification.

Another factor that can influence classification is the length of the clips. For this reason, as explained earlier, we applied different clip lengths and compared their performances, the results of which are tabulated in Table 4. When reviewing the results of the training and classification conducted by gradually reducing the clip length from 5 s, differences in the outcomes become apparent. The results indicated that the performance of the classifier was more robust and accurate when using longer audio clips, which allowed us to overlook some of the short momentary errors.

In T02, there is a shift in the results between clip lengths of 4 s and 3 s, and in T03, between clip lengths of 3 s and 2 s. Nevertheless, the differences are very small at 0.104% and 0.306%, respectively, and can be considered negligible. On the other hand, shorter audio clips may improve the temporal precision of water-activity detection. Considering these, we decided to set the clip length to 3 s in this study.

#### 3.3.3. Data Size for W-YAMNet Training

As mentioned earlier, sounds generated from the same actions, for example, a shower, inherently share common characteristics. However, due to the distinct environmental features of each bathroom, creating a classifier that can be universally applied to all bathrooms is almost impossible. Consequently, a model trained with data collected from edge devices is suitable only for the specific bathroom where the data was acquired. In this context, shorter data collection times, meaning less data, are advantageous for practical applications, as data collection and processing occur in the bathroom currently in use by the residents.

From this perspective, we examined the relationship between the amount of data used for training W-YAMNet and its performance. The experiments were conducted in three scenarios: one where all the collected data were used for training, another where 2/3 of the data were used, and, lastly, one where only 1/3 of the data were used. We evaluated the classification performance in each case with a clip length of 5 s and tabulated the results in Table 5. As expected, the best results were observed when employing all the available training data, and with a reduction in the data volume, the performance gradually declined. However, whether at the frame level or clip level, using two-thirds of the data did not lead to a substantial reduction in performance in comparison to the complete dataset. While these observations may not be based on a rigorous analysis, they suggest that reducing the amount of data during W-YAMNet training to this extent can still lead to satisfactory performance, offering increased convenience in practical applications.

### 3.4. Real-Time Application

This experiment involves deploying the developed classifiers in each bathroom to assess their real-time operational performance. Following a scenario designed to mimic actual bathroom usage, real users generated various sounds in the bathroom. The classifiers then identified and categorized these sounds, and the results, along with their timestamps, were stored on a cloud-based computer. The classifiers were stored on an edge computer based on Raspberry Pi 4B, as explained in Section 3.1. Considering the results from Section 3.2 and Section 3.3, the clip length for all audio data was set to 3 s. In other words, an audio clip was generated every 3 s, and this clip was fed into YAMNet to determine whether it contained water sounds. A threshold of 0.25 was set, so that if the YAMNet output exceeded this value, the clip was further processed by W-YAMNet. Ultimately, the outputs of both YAMNet and W-YAMNet were used to determine the specific activities taking place or if there were none, and the results and log data were stored on the Firebase server.

The scenario involved conducting approximately 10 min of continuous recording in each bathroom. During this period, two instances of FL, two instances of FA, and one instance of SH were performed at random times, and their sounds were recorded. Additionally, various natural sounds, such as footsteps, electric razors, hairdryers, conversations, and music, were introduced at arbitrary times with random durations. These sounds occasionally overlapped with water-related sounds and sometimes occurred during periods without water-related sounds. Here, actual residents simulated the water usage, and this data was input into the edge computer, with only the log data of the estimated results sent to the cloud computer. In real-world applications, labeling is not necessary; however, in this case, users manually wrote and submitted water-usage records for performance evaluation.

Figure 6 represents the performance results in each bathroom. In the figure, there are two horizontal bands for each bathroom. The upper band labeled “ANNO” represents the actual activities labeled with colors; for example, FL is represented in red. The lower band labeled “PRED” represents the classification results of the developed classifiers. In this representation, gray is defined as “Others”, meaning that W-YAMNet did not operate due to a low WRS score value output from YAMNet. Input sounds other than water-related sounds mentioned above are not shown in the “ANNO” data.

Upon examining the results, we can observe a high number of false negative errors, where water sounds were classified as non-water sounds, while false positive errors, where non-water sounds were classified as water sounds, were very low. False negatives were primarily associated with instances where external sounds entered an unoccupied bathroom or when non-water-related sounds were blended with water sounds during use. Nevertheless, the system’s inability to detect FL around the 140-s mark in T03 is an unusual occurrence, given that FL recognition errors are generally minimal. The fact that SH was identified as FL near the 230-s mark in the same bathroom further illustrates that the developed classifier is not infallible.

## 4. Discussion

A distinctive feature of the proposed system is its two-fold application: initially utilizing the publicly available YAMNet model, trained on extensive acoustic data, to preliminarily identify the presence of water sounds, and subsequently applying the W-YAMNet model—specially trained through transfer learning to distinguish between FLUSH, SHOWER, and FAUCET sounds—for sounds deemed water-related.

As observed in Section 3.2, we determined whether the input sound was a water sound using the output of YAMNet. Due to the two-stage classification approach, we could bypass the training procedure for non-water sounds, eliminating the need to address complex-to-train classes such as “Others”. On the other hand, the threshold value was obtained through trial and error; however, it would be better to discover a statistical and systematic method for this.

On the audio clip length, audio clips longer than 3 s demonstrated satisfactory classification performances. Given that water-usage sounds are not singular events but exhibit stationary characteristics with recurring statistical features, it is posited that YAMNet’s fundamental input unit—a frame of 0.96 s—serves as an appropriate basic unit for timbre evaluation. Furthermore, we determined that clip-level classification is more appropriate than frame-level classification, and this was validated through experiments. Once the clip length was fixed, we set the frame duration to 1 s, considering the standard frame size in YAMNet input, and ensured that adjacent frames had a 50% overlap. Through experiments, we concluded that 3 s is an appropriate duration [22].

This study presents a few limitations and challenges. Primarily, the parameters of the classifier layer retrained through transfer learning do not operate as global classifiers. Instead, the model requires training data from the home where the device will be installed due to significant differences in sounds and user habits. The challenge lies in obtaining diverse training datasets. Although requiring data acquisition for every application is a drawback, this structure was chosen to ensure classification performance. In addition, the amount of data needed for training varies by activity, and further studies are needed. Secondly, before implementing W-YAMNet, sounds are input into YAMNet to discern if they result from water usage. Looking at the comprehensive evaluation results in Section 3.4, it is evident that the reduction of false positives has been successful, but false negatives occur relatively frequently. Therefore, there is a need for post-processing methods to improve their accuracy in this regard. The sensitivity and specificity at this stage are not sufficiently high, leading to the error levels detailed in Section 3.1. This method can be expanded to detect more varied indoor daily activities [23]. Moreover, the approach can potentially be extended to detect emergency situations like falls through sounds or shouts for assistance, further assisting in ensuring the safety of solitary elderly residents.

The sound-based method as suggested cannot be considered the best solution in every area. A non-acoustic sensor would be more useful and effective in diverse scenarios. For instance, smart plugs available in the market would be useful for monitoring power-usage patterns. Alternatively, video-based methods would be most effective for monitoring activities in a living room. However, for monitoring activities in the bathrooms, a method based on sound analysis is deemed appropriate.

## 5. Conclusions

This paper proposes a method for classifying three bathroom water-usage activities: SHOWER, FLUSH, and FAUCET, soley based on sound. To address privacy concerns, an edge computer is placed in the bathroom to perform on-site sound detection and classification. Only the results and timestamps were transmitted to the cloud computer. The suggested method consists of two stages. First, YAMNet, a widely used sound classification network, is used to determine the presence of water sounds. Then, W-YAMNet is employed, which is trained to classify detailed water-usage activities while reflecting the acoustic characteristics of specific bathrooms. Through various experiments, suitable threshold values and clip lengths were determined, and an analysis was conducted on the amount of data required for training W-YAMNet. Applying this to continuous sound data from each bathroom has demonstrated its real-time usability, although there is still some room for improvement.

The proposed method allows for activity estimation based on sound in settings where surveillance cameras are impractical, providing the potential for broader applications across indoor activities. This technology holds promise for assessing the well-being and safety of independently living elderly individuals, enabling the analysis of their daily routines. Moreover, this technology can be integrated into automatic systems for swift responses during emergency situations, such as promptly attending to distress sounds or shouts for help from the elderly.

## Figures and Tables

**Figure 1 sensors-24-00022-f001:**
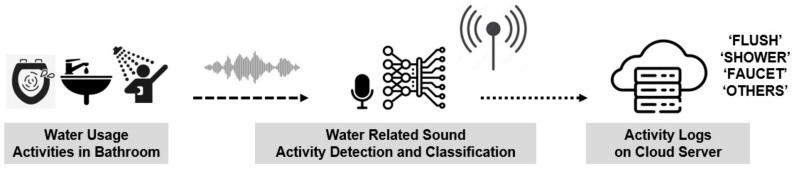
System configuration of the detection of daily activities based on sound-event classification.

**Figure 2 sensors-24-00022-f002:**
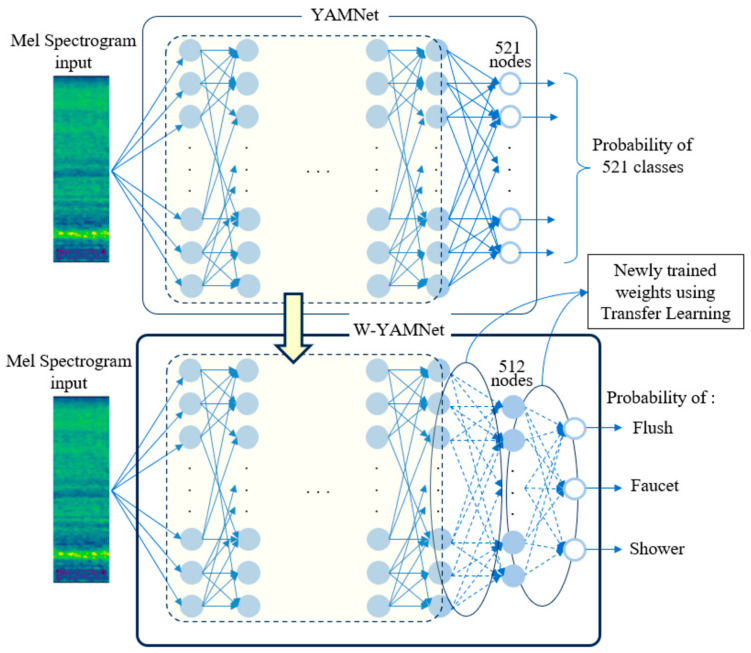
Construction of W-YAMNet from YAMNet.

**Figure 3 sensors-24-00022-f003:**
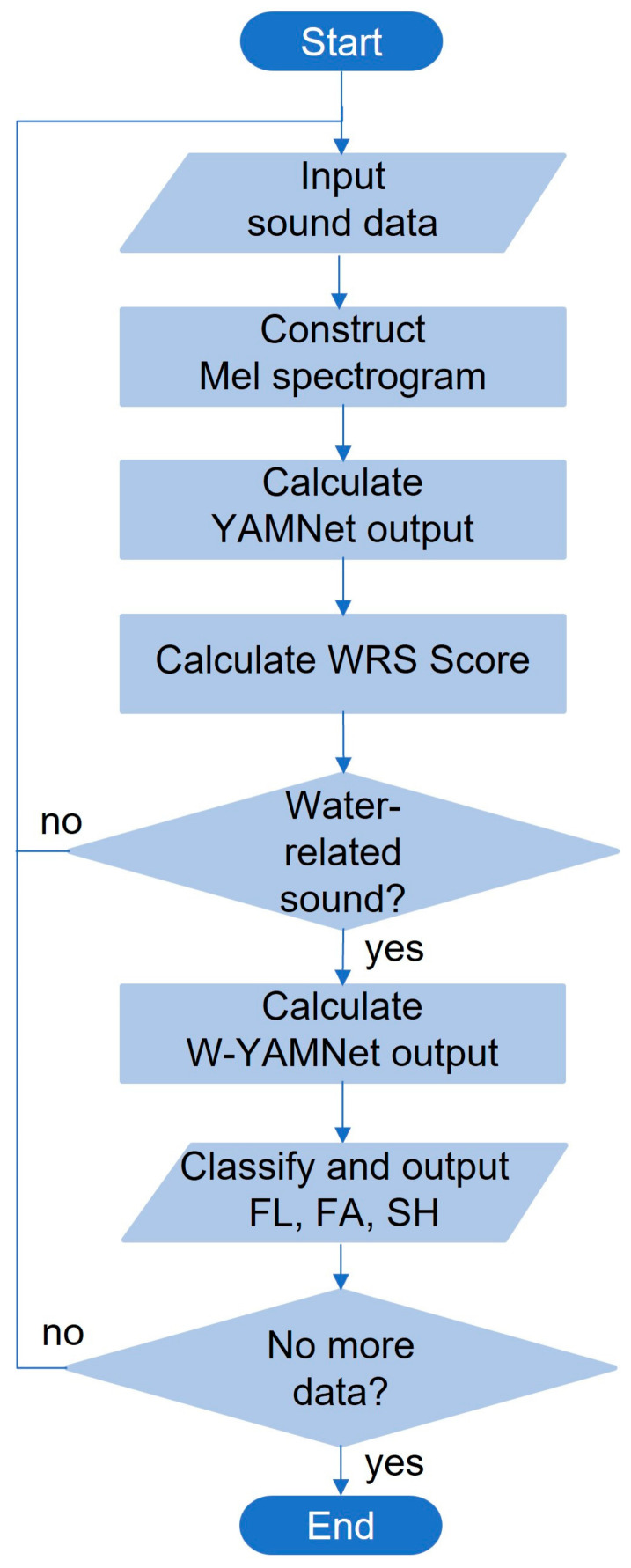
Flow chart of the proposed classifier.

**Figure 4 sensors-24-00022-f004:**
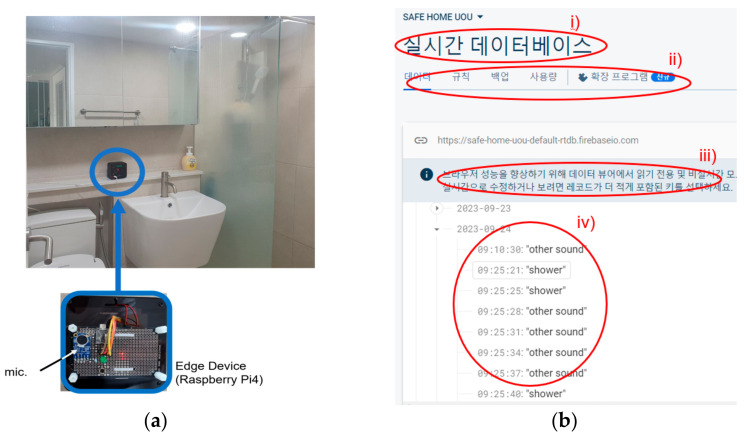
Implementation example. (**a**) Edge device with microphone in a recording bathroom. (**b**) Logging of daily activities onto the cloud server. In this figure, Korean texts are included that can be translated as: (i) the title of this page (written as Real-Time Database), (ii) menu of this page, (iii) brief description and (iv) log data (time and classification results).

**Figure 5 sensors-24-00022-f005:**
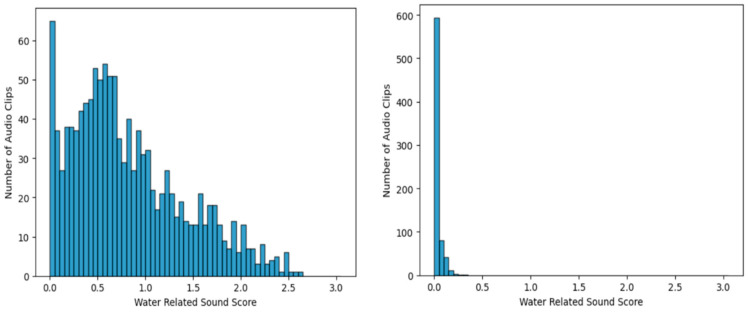
Histogram of water-related sound scores for the sound from water-usage activities (**left**) and from non-water-usage activities (**right**) from three distinct bathrooms.

**Figure 6 sensors-24-00022-f006:**
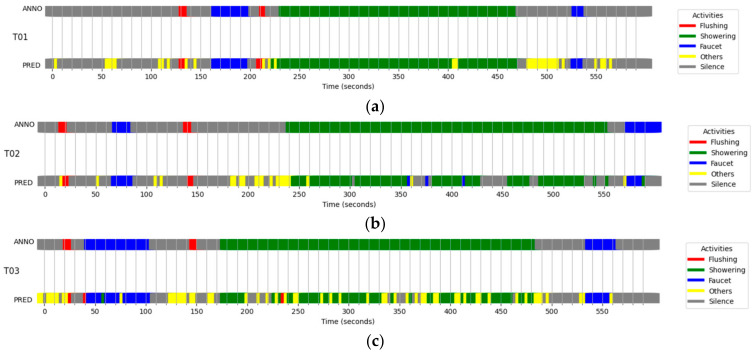
Ten-minute tests for the three restrooms, (**a**) T01, (**b**) T02, and (**c**) T03. Each data is composed of two rows of color strips. The upper strip is annotated with water-usage activities based on the recorded time, while the lower strip represents activities predicted by the W-YAMNet model. The yellow color labeled “Others” was not annotated in the upper strip.

**Table 1 sensors-24-00022-t001:** Top 10 classes of YAMNet output scores for the water-usage activity sounds in the bathroom.

Rank	Class Name	YAM Output	Class Index ^1^
1	Water ^2^	0.2001	282
2	Water tap, faucet	0.1788	364
3	Sink (filling or washing)	0.1434	365
4	Toilet flush	0.1088	368
5	Inside, small room	0.0762	500
6	Bathtub (filling or washing)	0.0637	366
7	Liquid	0.0585	438
8	Hiss	0.0474	79
9	Fill (with liquid)	0.0471	776
10	Steam	0.0441	290
16	Drip ^3^	0.0205	442

^1^ Index in AudioSet Ontology. ^2^ Category class of sound from the motion of liquid. ^3^ In the ranking from 11 to 20, the “Drip” class is listed because it is a water-related sound.

**Table 2 sensors-24-00022-t002:** Datasets from three bathrooms for the training and evaluation of W-YAMNet.

	Bathroom ID	T01	T02	T03
Datasets				
Training (frames, 0.96 s)		850	780	1340
Validation (frames, 0.96 s)		280	260	440
Test (audio clips, 5 s)		60	102	78

**Table 3 sensors-24-00022-t003:** Comparison of F1 scores between clip-level and frame-level classification results based on clip length.

Decision Interval	Decision Unit	F1 Score		
		T01	T02	T03
5 s	Frame level	0.950	0.900	0.918
	Clip level	1.000	0.971	1.000
4 s	Frame level	0.953	0.893	0.917
	Clip level	1.000	0.952	0.990
3 s	Frame level	0.943	0.896	0.919
	Clip level	1.000	0.953	0.977
2 s	Frame level	0.939	0.880	0.899
	Clip level	0.993	0.937	0.980
1 s	Frame level	0.916	0.853	0.856
	Clip level	0.964	0.902	0.935

**Table 4 sensors-24-00022-t004:** Comparison of F1 scores and confusion matrices for various clip lengths.

Decision Interval		T01			T02			T03		
5 s		FL	SH	FA	FL	SH	FA	FL	SH	FA
confusion matrix	FL	18	0	0	34	0	0	22	0	0
	SH	0	21	0	0	34	0	0	30	0
	FA	0	0	21	1	1	32	0	0	26
F1 SCORE		1			0.980			1		
4 s		FL	SH	FA	FL	SH	FA	FL	SH	FA
confusion matrix	FL	22	0	0	42	0	0	28	0	0
	SH	0	26	0	0	38	4	0	37	1
	FA	0	0	26	0	2	40	0	0	33
F1 SCORE		1			0.952			0.99		
3 s		FL	SH	FA	FL	SH	FA	FL	SH	FA
confusion matrix	FL	30	0	0	57	0	0	37	0	0
	SH	0	35	0	0	50	6	0	49	2
	FA	0	0	35	1	1	54	1	0	43
F1 SCORE		1			0.953			0.977		
2 s		FL	SH	FA	FL	SH	FA	FL	SH	FA
confusion matrix	FL	45	0	0	85	0	0	56	0	0
	SH	0	53	0	1	74	10	0	73	3
	FA	0	1	52	3	2	80	1	0	66
F1 SCORE		0.993			0.937			0.98		
1 s		FL	SH	FA	FL	SH	FA	FL	SH	FA
confusion matrix	FL	90	0	0	169	0	2	111	0	2
	SH	2	101	4	1	140	29	3	140	10
	FA	0	5	101	11	7	152	10	1	123
F1 SCORE		0.964			0.902			0.935		

**Table 5 sensors-24-00022-t005:** Comparison of F1 scores and confusion matrices for various data sizes (with a clip length of 5 s).

Used Data		T01			T02			T03		
Full		FL	SH	FA	FL	SH	FA	FL	SH	FA
confusion	FL	18	0	0	34	0	0	22	0	0
	SH	0	21	0	0	34	0	0	30	0
	FA	0	0	21	1	1	32	0	0	26
F1 SCORE		1			0.98			1		
2/3		FL	SH	FA	FL	SH	FA	FL	SH	FA
confusion	FL	18	0	0	34	0	0	22	0	0
	SH	0	21	0	0	28	6	0	30	0
	FA	0	0	21	1	1	32	0	0	26
F1 SCORE		1			0.921			1		
1/3		FL	SH	FA	FL	SH	FA	FL	SH	FA
confusion	FL	18	0	0	34	0	0	22	0	0
	SH	0	21	0	0	24	10	0	30	0
	FA	0	0	21	0	0	34	2	0	24
F1 SCORE		1			0.853			0.919		

## Data Availability

Data are contained within the article.

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
