# Peer review of "Sound-Event Detection of Water-Usage Activities Using Transfer Learning"

_sensors, 2023, doi:10.3390/s24010022_

Round 1

Reviewer 1 Report

Comments and Suggestions for Authors

The paper is devoted to a signal processing algorithm than converts conventional mini-PC with a microphone into a water usage sensor for apartments. The key exploited technologies are YAMNet model, pretrained (by other researchers) using data from Youtube, and transfer learning method to achieve higher prediction accuracy for a custom problem.

The paper is well-written. Instruments and methods are carefully described. References to recent studies are present. However, there are some problems in the text:

1) There are several issues in section 3.2. The first paragraph goes twice. The references to Table 1 are in fact for Table 2. Table 1 and Table 2 captions are mixed up.

2) One more thing about Table 1, I mean the presnt Table 1, with “Bathroom ID” in the top of it. “Bathroom ID” looks like a title for the first column, but in fact it is the title for the first line. It is not clear what the difference between “Validation” and “Test” is.

3) It is not clear how annotation has been done. Does the researcher, who pushed the button on flush wrote each time in a log? Or does the researcher listened for the recorded sounds afterwards to make labels on it?

4) It is not clear what was the ratio between water fragments and non-water fragments in training and testing datasets.

5) Unfortunately, I still don not get the actual need in this sensor. Almost all modern houses and apartments are facilitated with gauges that measure waterflow directly in tubes. Some ultrasonic sensors can be mounted on the tube, not in the tube, avoiding possible leaks. Available gauges are often equipped with electronic output for their data. E.g., they can send data to issue bills with exact price for the consumed water. The authors claim that they care about privacy of those, who are going to use their sensors. However non-acoustic sensor is more private and secure solution anyway. These facts should be discussed. But this remark does not prevent from publishing new scientific ideas.

Author Response

The author appreciate reviewer's valuable feedback.

Detailed responses are included in the attached file, please.

Reviewer 2 Report

Comments and Suggestions for Authors

The paper presents an interest topic related real application. And the method seems potential application as predicted in the context. Although,  it was always insufficient for more general scenario. Anyway,  this 2 - fold way worth considering.  

1. line 154 -155, For convenience, let's refer to the sum of these probabilities as the "Water-Related sound score (WRS score).", it was not so clear express the rule for scoring.

2. From line 179 to line 190,  it was discussed about the structure of W-YAMNet , and shown in figure 2.  The results vary from probability of 512 nodes to 3 classes  events.  It constrains results to 3 kinds, how to deal with other event happens in water scenario? 

3. For the Experimental Results, it seems all data captured were conducted in bathrooms, it was querying that how about that happened in more general scenario how about results will be.

Comments on the Quality of English Language

Minor English language revise can make this paper more fluent.

Author Response

The author appreciates the reviewer's valuable feedback.

Detailed responses are included in the attached file.
